# Engineering new-to-nature biochemical conversions by combining fermentative metabolism with respiratory modules

Helena Schulz-Mirbach [1,2,7], Jan Lukas Krüsemann[1,2,3,7], Theofania Andreadaki[2], Jana Natalie Nerlich[3], Eleni Mavrothalassiti[2], Simon Boecker [4,5], Philipp Schneider[4], Moritz Weresow[2], Omar Abdelwahab[3], Nicole Paczia [1], Beau Dronsella [1,2], Tobias J. Erb [1,6], Arren Bar-Even [2,8], Steffen Klamt [4] & Steffen N. Lindner [2,3] ✉

Anaerobic microbial fermentations provide high product yields and are a cornerstone of industrial bio-based processes. However, the need for redox balancing limits the array of fermentable substrate-product combinations. To overcome this limitation, here we design an aerobic fermentative metabolism that allows the introduction of selected respiratory modules. These can use oxygen to re-balance otherwise unbalanced fermentations, hence achieving controlled respiro-fermentative growth. Following this design, we engineer and characterize an obligate fermentative *Escherichia coli* strain that aerobically ferments glucose to stoichiometric amounts of lactate. We then re-integrate the quinone-dependent glycerol 3-phosphate dehydrogenase and demonstrate glycerol fermentation to lactate while selectively transferring the surplus of electrons to the respiratory chain. To showcase the potential of this fermentation mode, we direct fermentative flux from glycerol towards isobutanol production. In summary, our design permits using oxygen to selectively re-balance fermentations. This concept is an advance freeing highly efficient microbial fermentation from the limitations imposed by traditional redox balancing.

Triggering the metabolic switch between cellular respiration and fermentation in microorganisms served humans for centuries: far before knowing about microbial metabolism, humans have already unknowingly employed it for the production of fermented goods like yogurt, kimchi, bread, cheese and alcoholic beverages[1]. Creating micro- or anaerobic environments as a simple process engineering step did not require understanding of the underlying metabolic principles for the biotechnological use of microbes. Coherently, the terms food fermentations and industrial fermentation are broadly phrased and refer to the microbial conversion of raw foods or microbial processes that yield commercial products like biomass, proteins, or primary and secondary metabolites[2], respectively. However, throughout the past 200 years, the mechanism and types of fermentative metabolism were elucidated and are now well-understood and regularly used in biotechnology. Here, the

[1]Max Planck Institute for Terrestrial Microbiology, Karl-von-Frisch-Str. 10, 35043 Marburg, Germany. [2]Max Planck Institute of Molecular Plant Physiology, Am Mühlenberg 1, 14476 Potsdam-Golm, Germany. [3]Department of Biochemistry, Charité Universitätsmedizin Berlin, corporate member of Freie Universität Berlin and Humboldt-Universität, Charitéplatz 1, 10117 Berlin, Germany. [4]Max Planck Institute for Dynamics of Complex Technical Systems, Sandtorstraße 1, 39106 Magdeburg, Germany. [5]Berliner Hochschule für Technik (BHT), Seestr. 64, 13347 Berlin, Germany. [6]Center for Synthetic Microbiology (SYNMIKRO), Karl-von-Frisch-Straße 14, 35043 Marburg, Germany. [7]These authors contributed equally: Helena Schulz-Mirbach, Jan Lukas Krüsemann. [8]Deceased: Arren Bar-Even. ✉e-mail: steffen.lindner@charite.de

biochemical definition of fermentation refers to metabolic processes where the secretion of reduced products allows redox balancing reducing equivalents generated by cellular catabolism. When grown aerobically, facultative anaerobes like *E. coli* couple redox balancing with ATP synthesis by using the available oxygen as a terminal electron acceptor for the electron transport chain (ETC = respiratory chain). In short, membrane-bound dehydrogenases transfer electrons from NAD(P)H generated during catabolism or directly from metabolites (e.g. succinate, malate, pyruvate, glycerol 3-phosphate and lactate) to the quinone (Q) pool. This results in the production of membranous quinol ($QH_2$), which in turn can be re-oxidized by terminal oxidases that reduce oxygen to water (Fig. 1A). The resulting free energy is used to pump protons across the periplasmic membrane to build up a proton gradient. ATPases then couple the backflow of protons to the phosphorylation of ADP to generate ATP (oxidative phosphorylation). In the absence of oxygen, alternative terminal electron acceptors (e.g. nitrate, fumarate, dimethyl sulfoxide or trimethylamine *N*-oxide) allow anaerobic operation of the ETC to support oxidative phosphorylation[3].

Without a terminal electron acceptor, cells cannot perform oxidative phosphorylation. Instead, they can only generate ATP via substrate-level phosphorylation, which yields substantially less ATP per fed carbon source. Hence, electrons in reducing equivalents generated during glycolysis (mainly NADH) need to be transferred to intracellular metabolites that are excreted as fermentation products. These excreted fermentation products include organic acids (e.g. lactate, acetate, formate, and succinate) and alcohols (e.g. ethanol, butanediol, and butanol)[4]. It is important to note that, in total, their redox state has to be equal to the redox state of the fed substrate (=redox-balanced). Therefore, instead of being converted into biomass, a large fraction of the fed carbon ends up in fermentation products. Due to growth limitations from lowered ATP generation, fermentative growth typically exhibits lower biomass yields and high metabolite secretion compared to respiratory growth.

While the excretion of fermentation products represents a loss of fed carbon substrate for the cell, it can be exploited to maximize yields of target products in biotechnological processes. If, under anaerobic

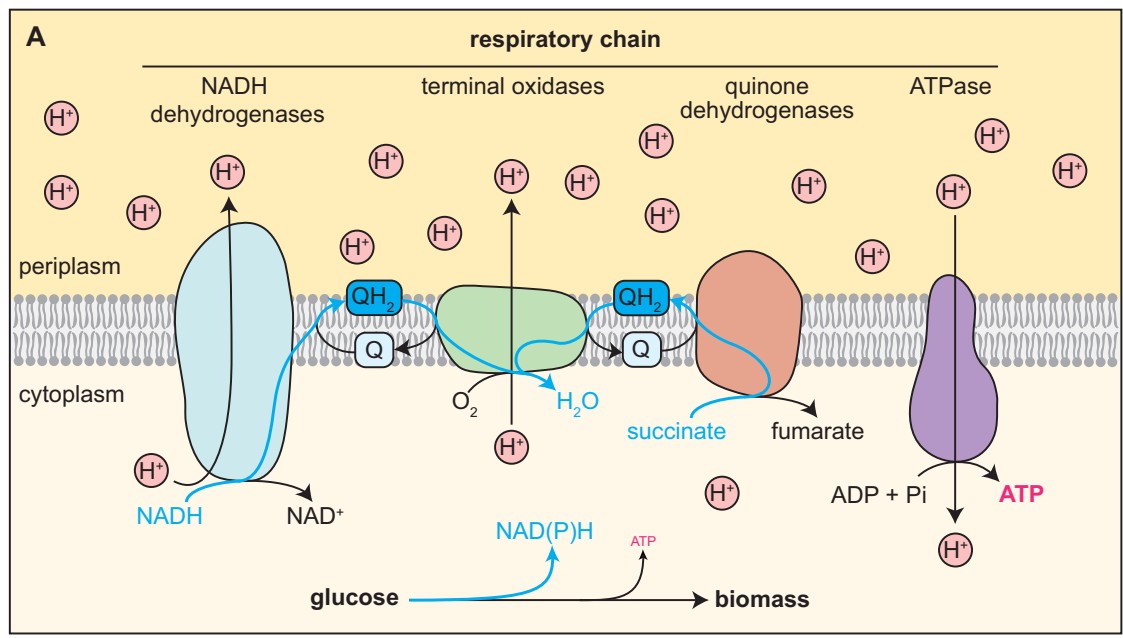

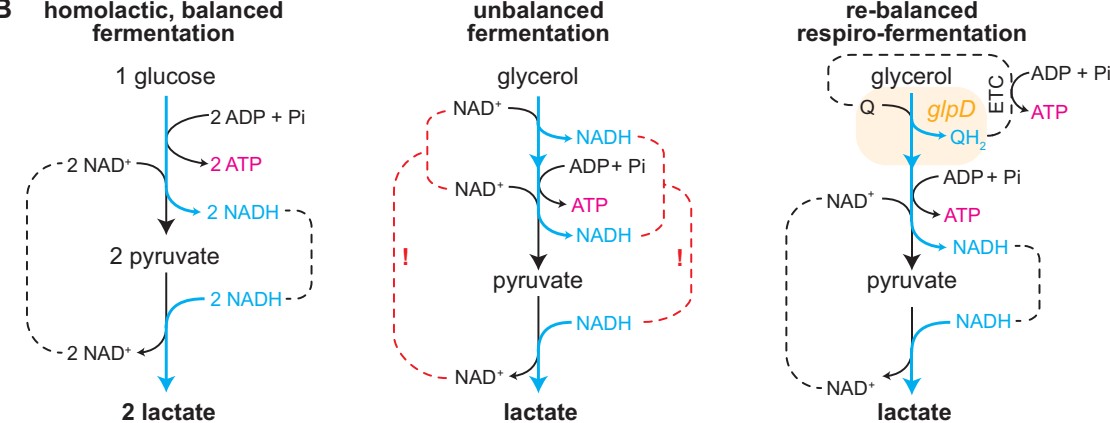

**Fig. 1 | Cellular electron flows during respiration and fermentation. A** Schematic representation of the electron transport chain. Electrons are transferred (blue arrows) from reducing equivalents to the NADH dehydrogenases (blue, Ndh and Nuo) and quinone dehydrogenases (orange, either NAD(P)H dependent ones or others, for example the succinate dehydrogenase). Quinones serve as intermediate electron carrier until oxygen serves as final electron acceptor for the terminal oxidases (green). Nuo and the terminal oxidases are generating a proton (H⁺) gradient which powers the ATPase (violet). This transfer allows quinone reduction to quinol; transfer through the respiratory chain is indicated by blue arrows. Abbreviations: H⁺, Q/$QH_2$−quinone/quinole. **B** Stoichiometry of homolactic fermentations. Left: balanced fermentation of glucose; middle: unbalanced fermentation of glycerol; right: re-balanced controlled respiro-fermentation of glycerol through the quinone-dependent glycerol 3-phosphate dehydrogenase (encoded by *glpD*).

conditions, undesired fermentation routes are eliminated to channel flux toward desired fermentation products, their growth-coupled production can be achieved[5–10]. This often results in higher product titers compared to respiratory production processes. Therefore, untapping the industrial potential of fermentative metabolism has been a prime target for metabolic engineering from the very beginning[11].

Several studies investigated to which extent fermentative behavior could be enforced under aerobic conditions. Disruption of the ETC and therefore respiratory metabolism was either achieved by the deletion of all terminal oxidases (cydAB, cyoABCD, and cbdAB)[12] or the main NADH dehydrogenases (encoded by the nuo operon and ndh) together with ubiquinone biosynthesis[13,14] (Fig. 1A). These studies yielded valuable insights into the mechanisms of an (aerobic) fermentative metabolism, which can not only be harnessed for bioproduction but also serve as a platform for the growth-coupled engineering of NADH consuming enzymes like oxygenases[13]. However, previously engineered aerobic fermentative strains were designed to be restricted to fermentative metabolism only and are therefore subject to the same metabolic limitations as non-engineered strains grown without terminal electron acceptors. This limits the feasible substrate/product combinations for fermentative production processes. Products like lactate or isobutanol can only be produced from substrates with the same degree of reduction like glucose (Fig. 1B). Thus, enforcing a strict aerobic fermentative metabolism does not represent an advantage over anaerobic fermentation processes.

In another line of research, supplementation with alternative electron acceptors or co-substrates was investigated to broaden the spectrum of substrate/product combinations allowing fermentative growth. Generally, such fermentations that would result in a reducing equivalent overaccumulation are coined unbalanced fermentations[15]. For example, the degree of reduction of glycerol would result in an unbalanced fermentation with lactate as a product (Fig. 1B, middle). Consequently, E. coli cannot grow on glycerol as a sole carbon source under strict anaerobic conditions[16]. Being a cheap by-product of biodiesel production[17], glycerol was the main target of engineering efforts aiming to re-balance its fermentation over the last decades[18–20]. The strategies allowing anaerobic growth on glycerol involved either addition of an external electron acceptor for redox balancing[16] or the co-feeding of complex additives like tryptone or yeast extract to support biomass formation[21,22]. Another strategy allowing anaerobic growth on glycerol was using an anode in a bioreactor as electron acceptor by the heterologous expression of electron transport proteins from Shewanella oneidensis[23]. While this electro-fermentation holds theoretical potential for simultaneous energy generation from the abstracted electrons, it is difficult to upscale and therefore less suited for industrial fermentation processes. Finally, the most commonly pursued strategy for fermentation of glycerol is the culturing under microaerobic conditions, which employ oxygen as electron acceptor[24]. Here, $O_2$ is supplied at limiting concentrations to allow redox balancing but not respiratory growth. Therefore, the cell is still forced to produce fermentation products for redox factor regeneration. While such altered process conditions might work well on a lab scale, industrial fermenters exhibit spatial gradients of $O_2$ distribution[25,26], which do not permit maintaining the required tight metabolic control.

In this work, we design and engineer a strain with an obligate aerobic fermentative metabolism that allows tightly controlled rebalanced fermentations. To achieve this, we leave the respiratory chain intact but delete all reactions transferring electrons to the quinone pool. The resulting E. coli strain grows under aerobic conditions on glucose in a homolactic fermentative manner. Next, we investigate to what extent our obligatory fermenting strain can be harnessed to rebalance otherwise unbalanced fermentations. We therefore reintegrate glycerol-3-phosphate:ubiquinone oxidoreductase (GlpD) as a respiratory module into the genome and thereby demonstrate rebalanced fermentation of glycerol to lactate. We call this growth phenotype controlled respiro-fermentative growth to reflect the combination of respiration and fermentation but to differ from natural, uncontrolled respiro-fermentative metabolism ("crabtree effect")[27,28]. Finally, to showcase the potential of controlled respiro-fermentative growth for industrially relevant processes, we replace the native lactate fermentation route with a heterologous pathway for isobutanol production. With this, we demonstrate the tremendous potential of our design as a platform strain enabling a broadly expanded array of feasible substrate/product combinations in microbial production processes.

## Results

### Eliminating electron transfer to the ETC results in an obligate fermentative strain

In our metabolic design, the ETC must stay intact to allow the functional re-integration of selected quinone-reducing reactions as respiratory modules. We therefore aimed to eliminate electron transfer to the ETC to prevent the use of oxygen as a terminal electron acceptor. For this, we identified all quinone-reducing reactions and their respective genes in the E. coli genome-scale model iML1515[29] and found 17 such reactions (Supplementary Data 1), which we manually curated (Supplementary Note 1). Briefly, we considered 6 reactions (ASPO3, DHORD2, DSBAO1, FDH4pp, HYD1pp, GlCDpp) not relevant and did not delete them to reduce the engineering effort while we added 3 reactions (lhgO, fadE and dadA) to the list that were not annotated as quinone-reducing in the model but in the EcoCyc database (Supplementary Note 1). The resulting list of 14 quinone-dependent reactions and their associated genes filtered to be relevant are shown in Fig. 2A. We identified seven potential NAD(P)H:quinone oxidoreductases (ndh, nuoABCDEFGHIJKLMN, kefF, wrbA, yieF, mdaB and its counterpart ygiN) and twelve quinone-dependent dehydrogenases including sdhABCD, lldD, poxB, fadE of which eight could potentially form a "mini-cycle" with an immediate NAD(P)H-dependent counterpart (dld, mqo, putA, glpD, glpABC, glcDEF, dadA, lhgO). Through these mini-cycles (e.g. ldhA and dld in Fig. 2), substrate cycling would allow NAD(P)H dependent quinone reduction and thus allow unwanted electron flow to the respiratory chain. As a first step, we deleted the main NADH dehydrogenases (Δndh ΔnuoEFG) and investigated the resulting phenotype. We found an impaired growth phenotype on different carbon sources (Supplementary Fig. 1A) as well as some aerobic lactate production (Supplementary Fig. 1B). However, after 50 h, the strain grew to high ODs associated with respiratory growth and consumed most of the initially produced lactate, which indicates that it could reactivate cellular respiration. Indeed, the specific glucose uptake rate during this growth phase was 8.42 mmol gCDW$^{-1}$ h$^{-1}$ (Supplementary Fig. 1C), which matches previous reports for the aerobic specific glucose uptake rate of a wild-type strain (7.62 ± 0.43 mmol gCDW$^{-1}$ h$^{-1}$)[30]. Upon quantifying the NADH/NAD$^+$ ratio of cells growing with 20 mM glucose, we found a slightly increased NADH/NAD$^+$ ratio for the Δndh ΔnuoEFG strain (Supplementary Fig. 1D–F), indicating metabolic perturbations by the deletion of the NADH dehydrogenases. To further push metabolism towards fermentative growth, we proceeded to delete the other identified genes. The resulting strain was called "NNmini" where "NN" denotes the deletion of the 2 main NADH dehydrogenases and "mini" denotes the elimination of all other quinone-reducing reactions including the ones potentially involved in mini-cycles (Fig. 2B).

We then characterized the growth phenotype of the NNmini strain. For this, we first performed an aerobic growth experiment where we compared the growth of a wild-type E. coli strain with the engineered NNmini strain on a gradient of glucose concentrations. Here, slower growth (wild-type growth rate on 20 mM glucose: 0.6502 ± 0.0052 h$^{-1}$; NNmini growth rate on 20 mM glucose: 0.0329 ± 0.0011 h$^{-1}$; Supplementary Fig. 2A, Fig. 3A) and lower biomass yields (Fig. 3B) were observed for the NNmini strain, which hinted at a

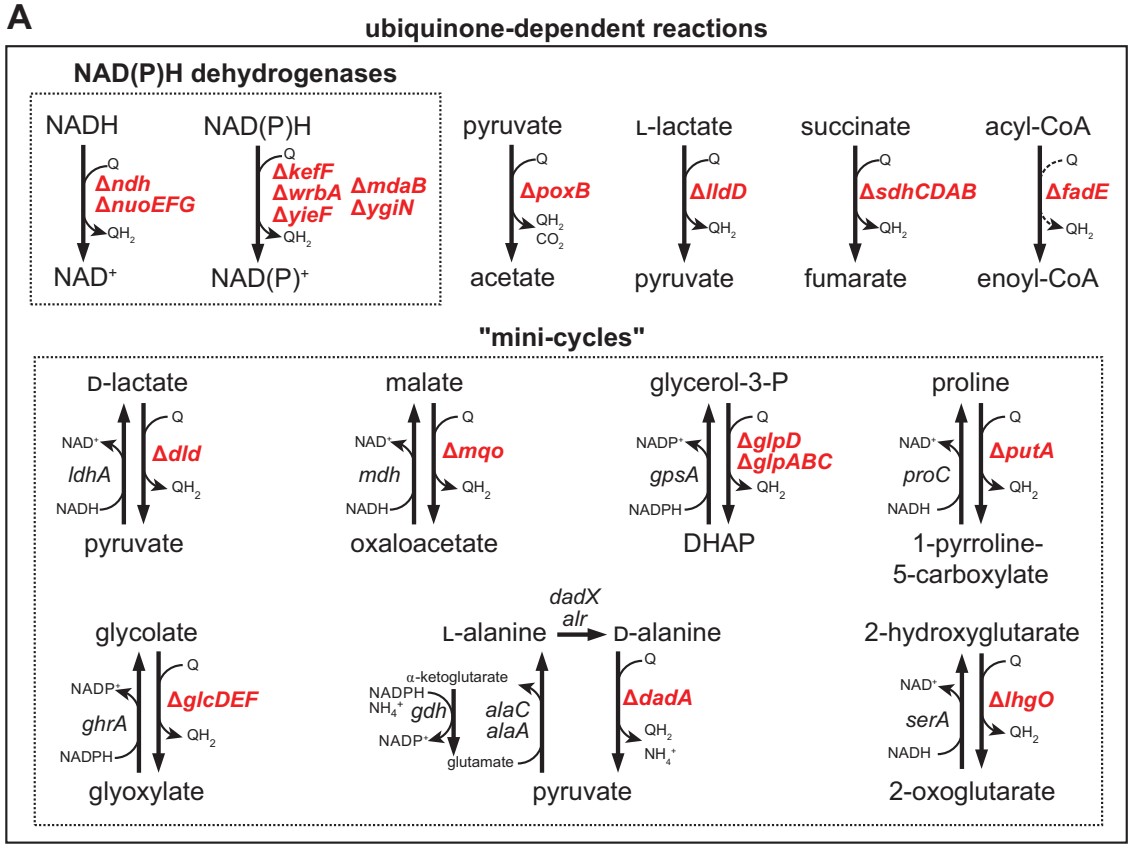

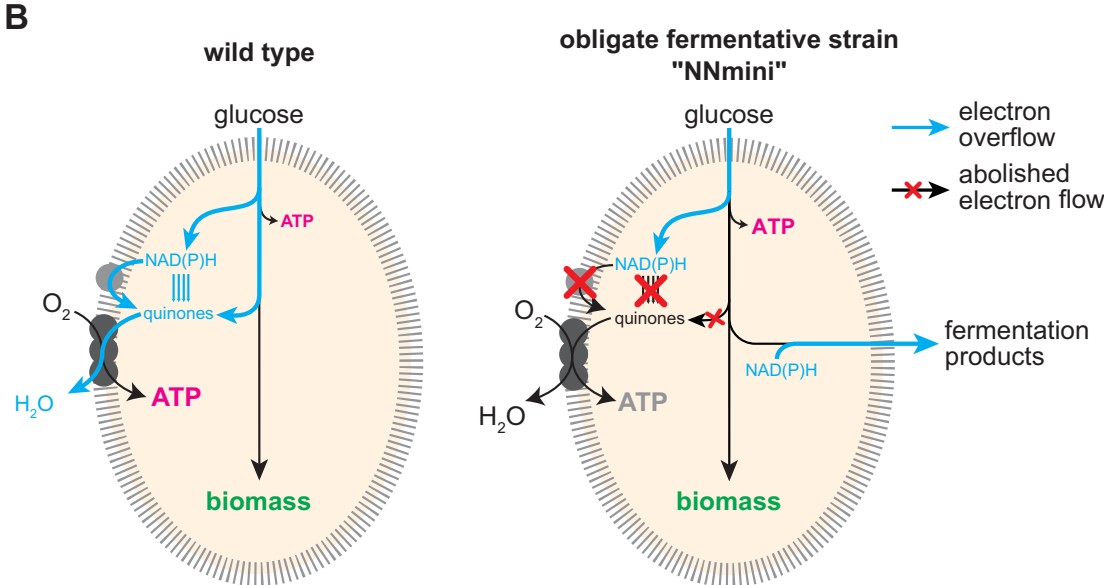

**Fig. 2 | Genetic interventions and concept of obligate fermentative NNmini strain. A** Known and putative ubiquinone dependent reactions were deleted. The top dotted box indicates genes encoding known (Nuo & Ndh) and putative NAD(P)H dehydrogenases and NAD(P)H:quinone oxidoreductases. In the case of the *nuo* operon (*nuoABCDEFGHIJKLMN*), deletion of *nuoEFG* is sufficient as the encoded subunits are essential for electron transfer and thereby essential for Nuo function. Note that in the iML1515 model, the reaction catalyzed by Nuo is denoted as a different reaction from Ndh due to the associated generation of a proton gradient.

Lower dotted box: Quinone-dependent dehydrogenases that can form mini-cycles with a direct NAD(P)H-dependent counterpart were deleted to prevent continuous electron transfer from reducing equivalents to quinones. **B** Schematic representation of electron fluxes (blue arrows) during cellular respiration in the wild type (left) and the aerobic fermentative NNmini strain (right). Due to disruptions of all reactions allowing electron transfer from reducing equivalents to quinones (indicated by red crosses), the strain relies on the secretion of fermentation products for redox balance maintenance.

fermentative growth phenotype. In line with this, the specific glucose uptake rate of the NNmini strain ($14.19 \pm 1.54$ mmol gCDW$^{-1}$ h$^{-1}$, Supplementary Fig. 2A) was comparable to the previously reported specific glucose uptake rate of a fermentatively growing wild type

($13.48 \pm 0.04$ mmol gCDW$^{-1}$ h$^{-1}$)[30]. We proceeded by quantifying metabolites from the culture supernatant and found that the NNmini strain converted fed glucose exclusively to lactate at a yield of $1.78 \pm 0.2$ mol lactate/mol glucose while no other fermentation

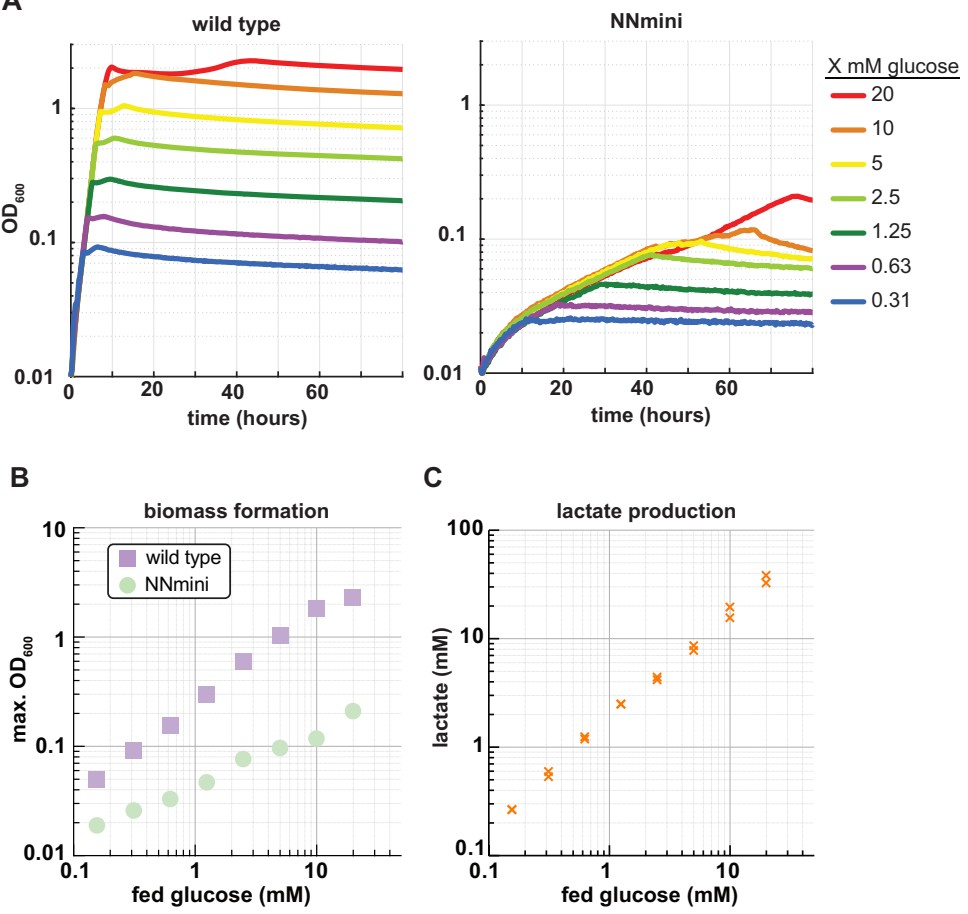

**Fig. 3 | Characterization of the NNmini strain in aerobic conditions. A** Growth of the NNmini strain (right) and a wild-type strain (left) on varying glucose concentrations. **B** The NNmini strain grows to much lower biomass yields, indicated by lower $OD_{600}$ values, than the wild type. The mean of duplicate measurements is shown. **C** Lactate concentrations detected in cultivation supernatants of the NNmini strain grown aerobically in minimal medium with varying glucose concentrations (yield $1.78 \pm 0.2$ mol lactate per mol glucose). For the wild type, no lactate was detected can therefore not be represented in a log scale. Duplicate measurements are shown. Source data are provided as a Source Data file.

product could be detected (Fig. 3C). This stands in stark contrast to the wild-type strain, where no fermentation product was detected. Furthermore, the NADH/NAD$^+$ ratio of the NNmini strain was increased ~16-fold compared to the wild type, which is in line with previous reports of ~10-fold[31,32] increased NADH/NAD$^+$ ratios of fermentatively grown wild-type cells compared to aerobically grown ones (Supplementary Fig. 2B–D). Thus, we concluded that elimination of quinone reducing reactions and by that, electron transfer to the ETC, resulted in an obligate fermentative phenotype.

Next, we investigated the anaerobic growth phenotype of both the wild type and our NNmini strain. For this, we repeated the glucose gradient growth experiment with the NNmini strain and a wild-type control under anaerobic conditions. Interestingly, we found that in anaerobic conditions, growth rates and biomass yields of the NNmini strain were closer to levels of the wild-type strain (wild-type growth rate on 20 mM glucose: $0.5095 \pm 0.0041$ h$^{-1}$; NNmini growth rate on 20 mM glucose: $0.4185 \pm 0.0033$ h$^{-1}$; Supplementary Fig. 3A). Furthermore, the NNmini strain exhibited higher growth rates and biomass yields than under aerobic conditions (Supplementary Fig. 3A). We quantified fermentation products and found that both strains exhibited hetero-fermentative behavior and excreted acetate, formate and ethanol in comparable amounts (Supplementary Fig. 3B). Taken together, these findings suggest that the observed homolactic fermentation of the NNmini strain under aerobic conditions is less efficient with regards to supported growth rates and biomass yields

compared to its native anaerobic fermentation mode. Indeed, the switch from respiration to fermentation is governed by the oxygen mediated expression control of ~200 genes though key regulators like the ArcAB system and FNR[33,34]. However, our metabolic engineering approach did not address this regulatory layer, which might result in the unnecessary expression of respiratory genes (and repression of important fermentative genes) in the NNmini strain, causing an increased metabolic burden[35], as well as insufficient metabolic control under aerobic conditions. However, whole-genome sequencing of the NNmini strain confirmed that the native regulatory network of the fermentative and respiratory metabolism was still intact, indicating that at least in our strain handling conditions, there was not enough selective pressure to evoke mutations concerning the regulatory network (Supplementary Data 2).

Finally, we investigated whether an engineering approach involving less genetic modifications could be employed to achieve an aerobic fermentative phenotype with an improved growth phenotype for easier handling compared to the NNmini strain. We hypothesized that removal of ubiquinone biosynthesis ($\Delta ubiCA$) in the $\Delta ndh$ $\Delta nuoEFG$ strain would result in a disrupted ETC and therefore an aerobic fermentative phenotype as it was previously observed[13]. We engineered the mentioned deletions and characterized the resulting strain (Supplementary Note 2, Supplementary Fig. 4). We found that although the strain exhibited fermentative growth and produced lactate aerobically, acetate was excreted as a byproduct. Furthermore,

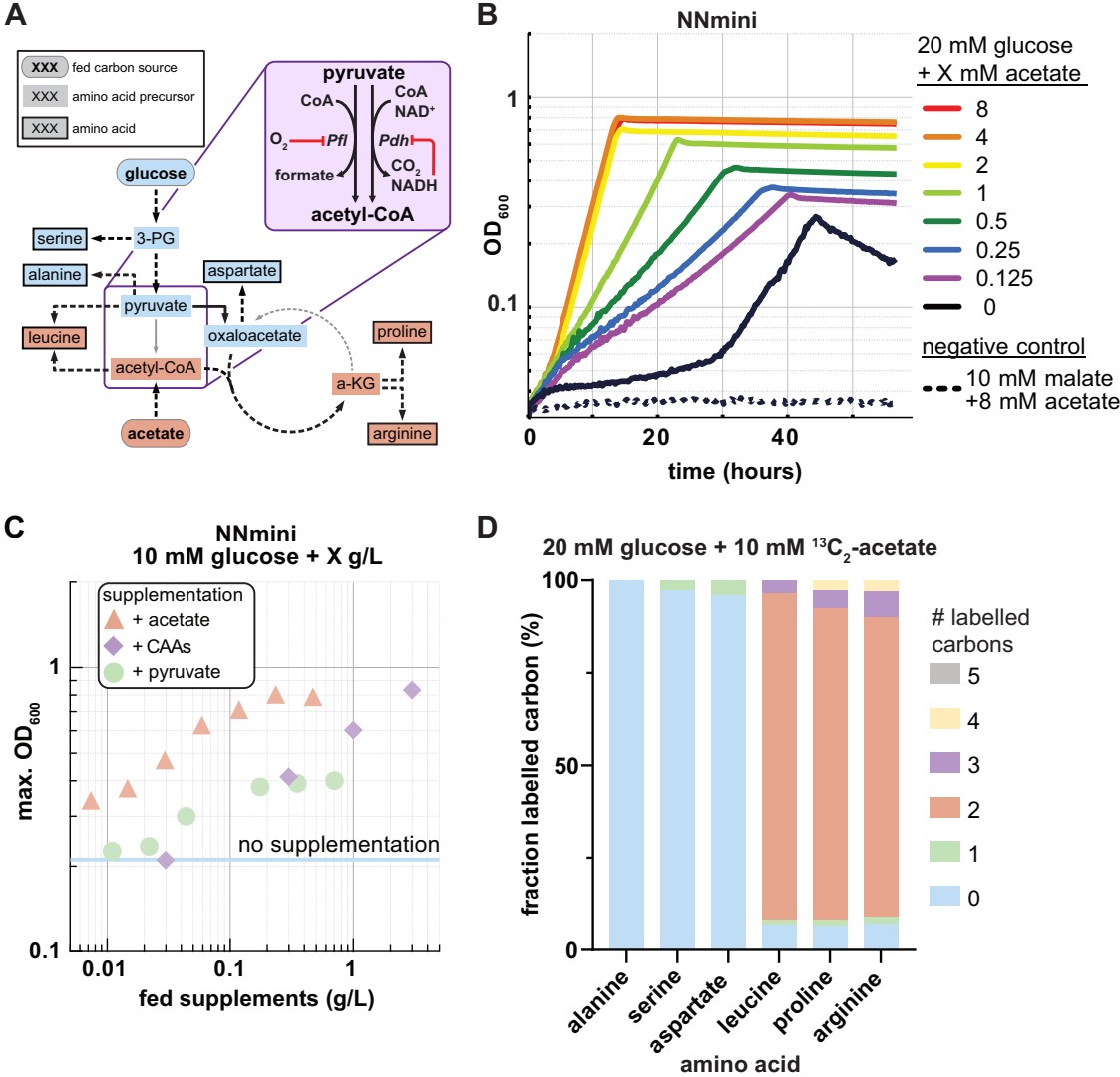

**Fig. 4 | Supplementation of acetate, pyruvate or casamino acids improve growth of the NNmini strain. A** Schematic view of expected carbon flow towards selected biomass precursors in NNmini strain grown on glucose with acetate supplementation. Since malate dehydrogenase (Mqo) and succinate dehydrogenase (Sdh) are absent, acetate can only be used for the generation of biomass precursors from lower metabolism. Carbons from acetate are therefore expected to be found in metabolites colored in orange. Of these, the amino acids leucine (made partly from pyruvate and acetyl-CoA), proline and arginine (both derived from a-KG) are expected to carry carbons from acetate, in contrast to amino acids from upper metabolism. Purple inset: Schematic view of the enzymes involved in the conversion of pyruvate into acetyl-CoA. Pfl is oxygen-sensitive and can therefore only operate under anaerobic conditions. Pdh produces NADH and is inhibited by it. **B** Growth of the NNmini strain on 10 mM glucose with different acetate concentrations. 20 mM malate (non-fermentable) + 10 mM acetate were used as negative control medium to demonstrate that acetate itself would not allow the NNmini strain to grow. **C** Comparison of the growth improvement from feeding different acetate, pyruvate or casamino acid (CAA) concentrations. Means of duplicates are shown, concentrations are given in g/L to allow comparison with CAAs, which are a complex medium without defined molecular weight. **D** Isotopic labeling patterns found in alanine, serine, aspartate, leucine, proline, and arginine obtained from the NNmini strain grown on 20 mM glucose + 10 mM $^{13}C_2$-acetate. Only acetyl-CoA derived leucine and a-KG derived proline and arginine are double-labeled. Abbreviations: a-KG α-ketoglutarate, 3-PG 3-phosphoglycerate, Pfl pyruvate formate lyase, Pdh pyruvate dehydrogenase. Source data are provided as a Source Data file.

upon the accumulation of nonsense mutations in the *ubiE* gene and a mutation in the *dld* promoter region, respiratory growth was re-established. Here, we hypothesized that an accumulation of demethylmenaquinone and the promiscuous use of it as electron acceptor by Dld potentially resulted in the disruption of the obligate fermentative phenotype. While such metabolic adaptation already is detrimental to any application of this strain, the inactivation of the ETC by the deletion of ubiquinone biosynthesis does not allow the reintegration of respiratory modules for unbalanced fermentations. Since the absence of quinone dependent dehydrogenases in the NNmini strain should make it robust to promiscuously used quinone species, we decided to further investigate the fermentative growth phenotype of the NNmini strain and how it could be improved.

## Acetate, pyruvate or casamino acid supplementation improves aerobic biomass yields of the obligate fermentative strain

Under aerobic conditions, the NNmini strain exhibited reduced biomass yields and growth rates compared to anaerobic growth. We hypothesized that this phenotype resulted from the inability to use pyruvate-formate lyase (Pfl) for acetyl-CoA formation during aerobic growth. While the oxygen sensitive pyruvate-formate lyase (Pfl), is active[36] under anaerobic conditions and allows the non-reductive cleavage of pyruvate to yield acetyl-CoA and formate[37], NADH-producing pyruvate dehydrogenase (Pdh) catalyzes the formation of acetyl-CoA from pyruvate under aerobic conditions (Fig. 4A). This has several implications: the NADH production from Pdh interferes with the redox-balanced conversion of glucose to lactate. Under anaerobic

conditions, the main alcohol dehydrogenase AdhE of *E. coli* can alleviate this redox imbalance through ethanol production as an electron sink, however, the enzyme it is not expressed under aerobic conditions[38]. Therefore, acetyl-CoA formation via the Pdh would result in an excess NADH, which cannot be recycled by fermentation product formation and would thus lead to redox imbalance. Additionally, Pdh is inhibited by high NADH concentrations, which are common under fermentative conditions, and might therefore produce acetyl-CoA inefficiently (Fig. 4A)[39]. Taken together, the absence of Pfl abolishes the strain's possibility to produce acetate to generate additional ATP per pyruvate. Pdh is inhibited under fermentative conditions and generates an NADH surplus that causes an acetyl-CoA limitation resulting in impaired growth under aerobic conditions of the NNmini strain.

In the following, we aimed to test whether the supplementation of biomass precursors from lower metabolism could complement the acetyl-CoA demand. To this end, we supplied the NNmini strain growing on 10 mM glucose with different concentrations of acetate, pyruvate or casamino acids (CAAs). All three supplements improved both the growth rate and the formation of biomass to varying extents (Fig. 4B, C & Supplementary Fig. 5A, B). In case of acetate supplementation, despite the cost of at least 1 ATP for acetyl-CoA formation, growth was significantly improved (maximum $OD_{600} = 0.8$ with 4 mM acetate, max. growth rate = 0.277 $h^{-1}$). This hints that acetyl-CoA production limits growth of the NNmini strain rather than ATP yields from substrate-level phosphorylation. In case of CAAs, their supplementation yields free amino acids from lower metabolism as they are obtained from acid hydrolysis of casein and contain all amino acids except tryptophan. We therefore expected a reduced dependence on inhibited Pdh in the NNmini strain. Indeed, maximum $OD_{600}$ values similar to a wild type growing on 10 mM glucose were observed for CAA supplementation ($OD_{600} = 1.825$, max. growth rate = 0.315 ± 0.014 $h^{-1}$). In case of pyruvate supplementation, converting the supplied pyruvate to acetyl-CoA via Pdh results in the formation of one NADH. However, converting 1 molecule of pyruvate to 1 molecule of lactate also consumes one NADH. Each additional conversion of supplied pyruvate to lactate should thus lower intracellular NADH levels, thus reducing Pdh inhibition. Then, a fraction of the supplied pyruvate can be used for acetyl-CoA synthesis via Pdh, which should improve growth at higher concentrations of supplied pyruvate. Indeed, biomass yields and growth rates of the NNmini strain supplemented with pyruvate were lower compared to acetate or CAAs (max. OD = 0.47 with 8 mM pyruvate, growth rate = 0.108 ± 0.003 $h^{-1}$). Taken together, our findings through supplementation with acetate, CAAs or pyruvate are in line with the speculated acetyl-CoA limitations of the NNmini strain due to NADH-dependent Pdh inhibition.

To further test our hypothesis and exclude that acetate somehow is assimilated into upstream metabolism we grew the NNmini strain on glucose in the presence of $^{13}C_2$-acetate and analyzed the isotopic labeling patterns in selected proteinogenic amino acids (Fig. 4A, D). To trace the labeled carbon throughout metabolism, we looked at alanine as a pyruvate-derived amino acid, serine from 3-phosphoglycerate and aspartate from oxaloacetate as example amino acids from upper metabolism precursors. Proline and arginine that are derived from α-ketoglutarate as well as leucine originating from pyruvate and acetyl-CoA were predicted to contain labeled carbons from fed $^{13}C_2$-acetate. Indeed, our analysis indicated that labeled carbon from acetate was only incorporated in leucine, proline and arginine (Fig. 4D). Thus, acetyl-CoA is indeed made from the supplied acetate and further incorporated into TCA cycle intermediates until α-ketoglutarate. From there, the deletions of *sdhCDAB* and *mqo* prevent further acetate utilization via the glyoxylate shunt, which is reflected in the absence of labeled carbon in alanine, serine and aspartate. Therefore, while the majority of cellular biomass is still made from the supplied glucose, using acetate for acetyl-CoA synthesis appears to be beneficial for the strain and could be considered in fermentation processes where high

microbial biomasses are required. Industrially, compared to pyruvate or CAAs, acetate is the best-suited supplement as it is cheap and can be produced from industrial waste streams[40].

## Integration of a respiratory module for the controlled respiro-fermentative conversion of glycerol to lactate

After engineering the obligate fermentative NNmini strain and phenotyping its growth, we turned our attention towards the goal of achieving "unbalanced" fermentations. As a first proof of principle, we aimed to demonstrate aerobic fermentation of glycerol to lactate (Fig. 1B). Being a byproduct of biodiesel production, glycerol is readily available and considered as a more renewable and sustainable carbon source than glucose. To make glycerol available for fermentative production processes, we present an approach which does not rely on any gene over-expression or process engineering adaptations. Since the NNmini strain still harbors an intact ETC, selected quinone-dependent reactions can be reintegrated, which allows the use of ubiquinone and thereby oxygen for single oxidation steps. Following this logic, reintegration of the quinone-dependent glycerol 3-phosphate dehydrogenase (GlpD) for the conversion of glycerol-phosphate to dihydroxyacetone-phosphate would allow maintaining the cellular redox balance while fermenting glycerol to lactate (Fig. 1B). The resulting hybrid growth mode combining fermentative growth with single respiratory modules will in the following be termed "controlled respiro-fermentation".

To test this design, we performed the genomic reintegration of *glpD*. While the base NNmini strain was unable to grow on glycerol as sole carbon source (Supplementary Fig. 6A), NNmini +*glpD* could aerobically grow on glycerol (growth rate with 40 mM glycerol: 0.126 ± 0.014 $h^{-1}$; Fig. 5B). Interestingly, the observed maximum $OD_{600}$ values were significantly higher compared to the aerobic fermentative growth on glucose (comparing 20 mM glucose with 40 mM glycerol to account for the number of carbons, Fig. 3A and Fig. 5B). This phenotype is likely due to respiratory ATP production in the ETC coming from the reoxidation of the quinol ($QH_2$) that is generated by GlpD (Fig. 1A). Measurements of intracellular ATP concentrations supported this hypothesis. Here, ATP concentrations of the NNmini + *glpD* strain grown on 40 mM glycerol were ~2.6 fold higher than those of the NNmini strain grown on 20 mM glucose (Supplementary Fig. 6B). Notably, ATP concentrations in both strains were still lower than in the Δ*nuo* Δ*ndh* strain grown on 20 mM glucose or 40 mM glycerol, respectively. Thus, ATP synthesis appears to be reduced (Supplementary Fig. 6B), which is in line with our expectation of reduced respiratory chain activity. Next, we quantified the lactate production by the NNmini + *glpD* strain and found that the supplied glycerol was converted to lactate in a nearly stoichiometric manner (Fig. 5D) with acetate as a minor byproduct for high glycerol concentrations (0.04 ± 0.03 mM acetate for 10 mM glycerol, 1.25 ± 0.05 mM acetate for 20 mM glycerol and 1.01 ± 0.92 mM acetate for 40 mM glycerol). Encouraged by these results, we aimed to extend the NNmini metabolic network by a non-native module to allow re-balanced glycerol fermentation to isobutanol.

## Respiro-fermentative isobutanol production from glycerol

Isobutanol production has been engineered and optimized in *E. coli*[41–44] as well as bulk production hosts like *Corynebacterium glutamicum*[45] and *Saccharomyces cerevisiae*[46]. Here, the formation of native fermentation products was abolished, which enforced the growth-coupled production of isobutanol as the sole electron sink for redox balance maintenance. However, all of these studies only focused on the balanced fermentation of isobutanol from glucose. Notably, fermentative isobutanol production from glycerol was demonstrated in *Klebsiella pneumoniae*[47]. However, the production was not achieved via growth-coupling with isobutanol as the only fermentation product since the engineered *K. pneumoniae* strain relied on its native 1,3-propanediol production for redox balance maintenance[48].

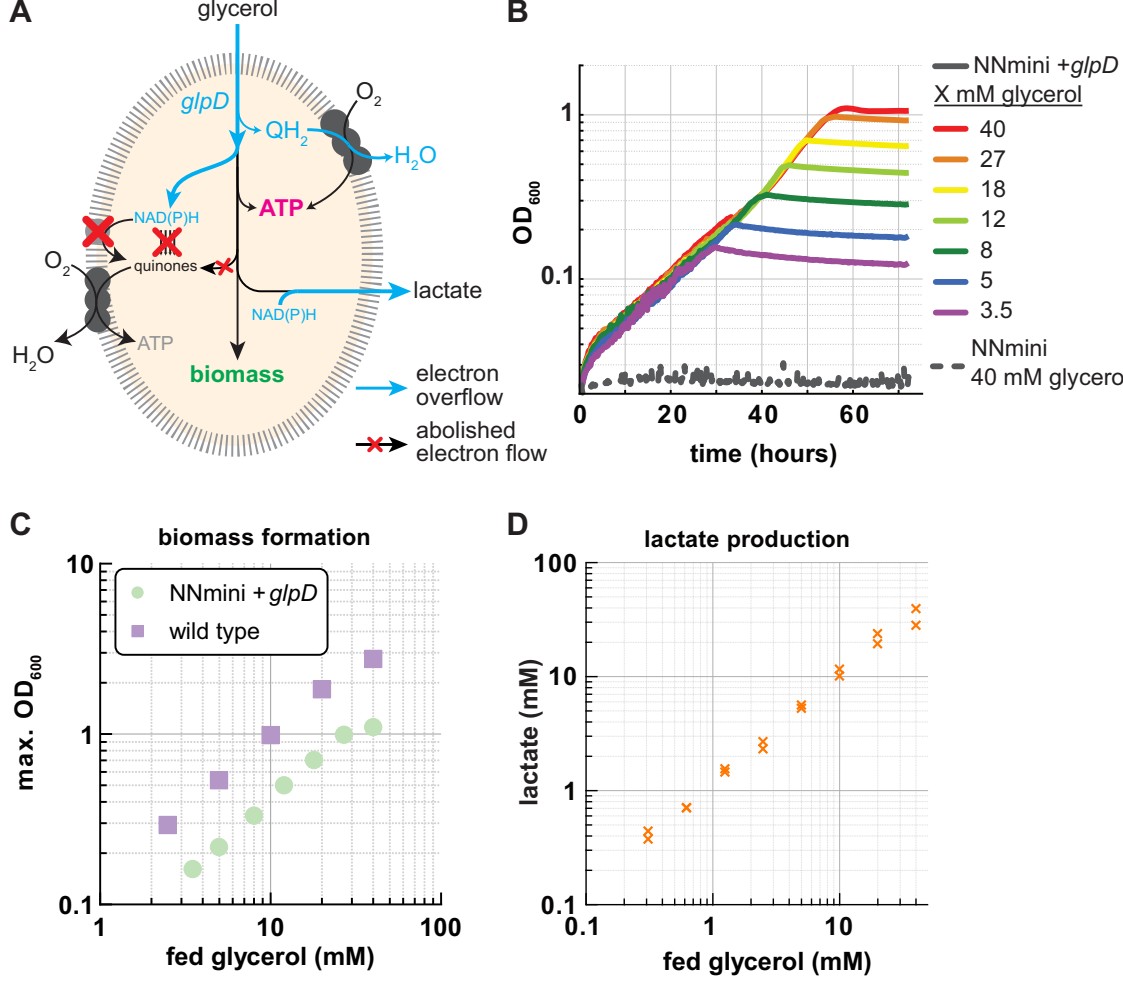

**Fig. 5 | Aerobic glycerol fermentation by the NNmini +glpD strain. A** Schematic representation of glycerol metabolism in the NNmini + *glpD* strain. Reintroduction of *glpD* allows transferring one electron pair to ubiquinone, which re-balances the fermentation of glycerol to lactate. **B** Growth of the engineered NNmini and the NNmini + *glpD* strain on minimal medium supplemented with varying glycerol concentrations (indicated to the right).

**C** Biomass yields, indicated by maximum $OD_{600}$ values, of the NNmini + *glpD* strain are lower than those of a wild type grown with glycerol as sole carbon source ($n = 1$). **D** Lactate detected in supernatants of the NNmini + *glpD* strain grown with varying glycerol concentrations (yield $0.85 \pm 0.14$ mol lactate/mol glycerol). Duplicate measurements are shown. Source data are provided as a Source Data file.

In this work, we aimed to use the NNmini strain for the controlled respiro-fermentative and thereby growth-coupled production of isobutanol from glycerol. To redirect the entire fermentative metabolism of the NNmini strain from lactate as main fermentation product to isobutanol, we introduced an isobutanol production plasmid (pIBA4, further on referred to as pIBA)[41] and deleted *ldhA* in the NNmini + *glpD* strain. Stoichiometrically, the production of isobutanol should be able to replace the native lactate fermentation while maintaining the cellular redox balance. Subsequently, we demonstrated that the NNmini Δ*ldhA* + *glpD* + pIBA could grow with glycerol as the sole carbon and energy source (Fig. 6B). As previously observed for lactate production, supplementation with acetate or casamino acids supported the growth of the strain (Supplementary Fig. 7A). Since the genes responsible for isobutanol production are under the control of an IPTG-inducible promoter on the pIBA plasmid, a condition without IPTG served as a negative control. While growth on glycerol could be achieved, the strain grew much slower compared to its lactate producing ancestor. This difference is potentially attributed to the associated protein and plasmid burden as the plasmid was optimized for a different genetic background[41]. In addition, redox factor recycling requires the activity of 3 heterologous (*alsS, kivd* and *adhA*) and 2 native genes (*ilvD* and *ilvC*) for isobutanol production compared to only 1 gene when lactate

is the fermentation product (*ldhA*). In the future, performing enzyme expression optimization as it was carried out in the original paper presenting the pIBA plasmid together with adaptive laboratory evolution could be used to improve pathway activity and thereby growth.

Next, we analyzed metabolites from the culture supernatants and found that glycerol was indeed converted to isobutanol (Fig. 6C, Supplementary Fig. 7B). While the observed titers are more than double the one reported for isobutanol production from glycerol in *K. pneumoniae*[47], these yields are 80% of the maximum yields ($2.012 \pm 0.059$ mM isobutanol from 5 mM glycerol): in theory, one isobutanol molecule could be produced from two glycerol molecules (Fig. 6A). Notably, the isobutanol yields achieved with the NNmini Δ*ldhA* + *glpD* + pIBA could potentially be negatively affected by the evaporation of the highly volatile isobutanol from the growth medium, which has previously been identified as problem in microbial isobutanol production[49]. Besides isobutanol, we also detected some ethanol in the growth medium. This might be due to the overexpression of *adhA*, which is part of the isobutanol biosynthesis pathway but has a side activity with acetaldehyde resulting in ethanol production[50]. Our findings demonstrate the aerobic growth-coupled production of isobutanol via controlled respiro-fermentative growth. This showcases that the concept of the NNmini strain frees

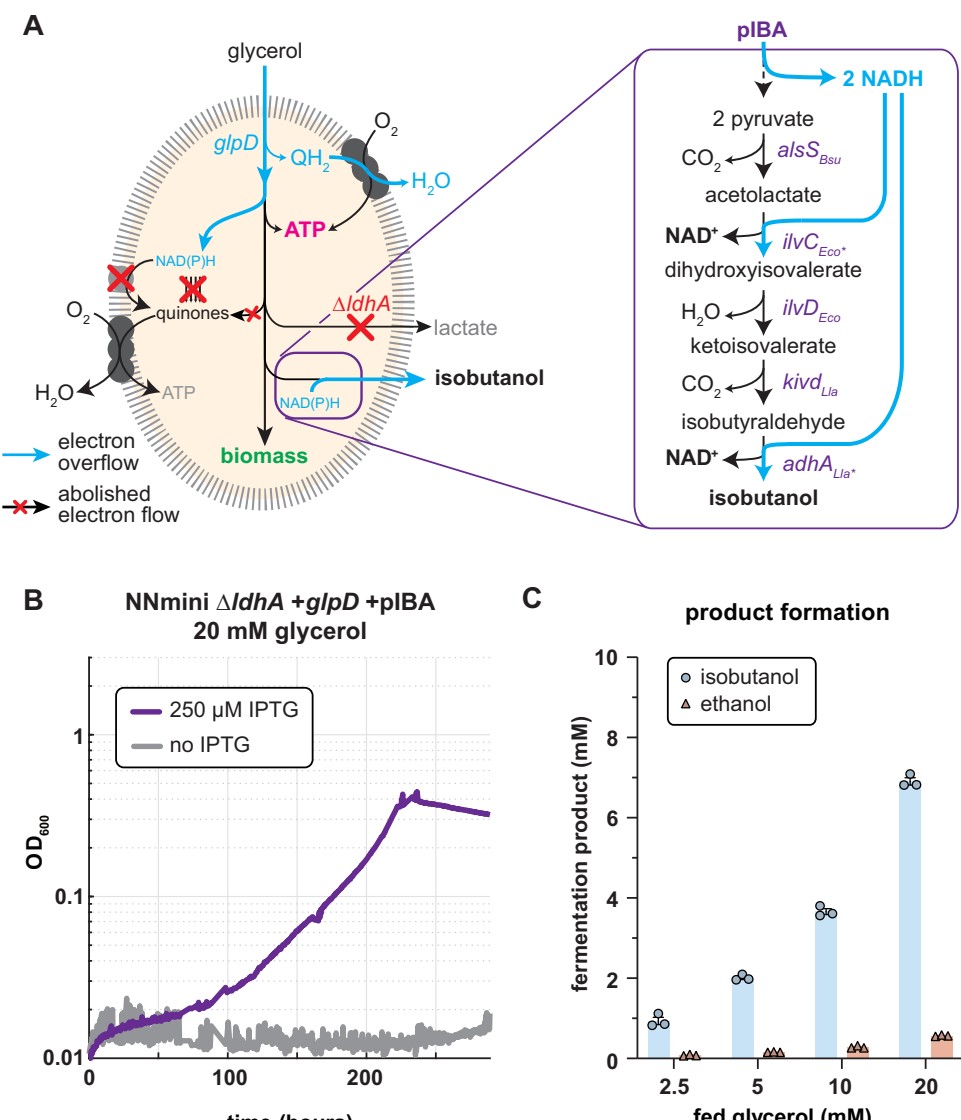

**Fig. 6 | Controlled respiro-fermentative glycerol conversion to isobutanol by the NNmini ΔldhA +*glpD* + pIBA strain. A** Schematic representation of glycerol fermentation to isobutanol. By deleting the native lactate synthesis route (Δ*ldhA*), the metabolic flux can be re-directed towards isobutanol. The genes for isobutanol synthesis are expressed from the pIBA plasmid (optimized pIBA4[41]). The isobutanol pathway scheme was adapted from Ghosh et al. [41]. **B** Growth of the NNmini Δ*ldhA* +*glpD* + pIBA strain grown with glycerol as sole carbon source with and without IPTG. Here, acetate served as a negative control in addition to the "no IPTG" condition. **C** Fermentation products detected in the supernatants of the NNmini Δ*ldhA* + *glp*D + pIBA strain grown with different glycerol concentrations. The mean values of triplicate measurements are shown with standard deviation. Source data are provided as a Source Data file.

fermentation from the limitation of requiring strictly redox-balanced substrate/product combinations.

## Discussion

Engineering obligate fermentation has so far relied on the disruption of the ETC, which was either achieved by deleting the terminal oxidases[12], or by deleting quinone biosynthesis[13,14] (see Supplementary Note 2 on Δ*ubiCA*, NNQ strains). Notably, these metabolic interventions can be understood as "flip-switches", which permanently change the growth mode of *E. coli* from respiration to fermentation. Since they exclude the use of oxygen as an electron acceptor, these approaches do not allow the combination of fermentation with respiratory modules (=controlled respiro-fermentation).

By deleting both NADH dehydrogenases and all quinone-reducing reactions, we flipped the switch from respiratory to fermentative growth and generated an obligate fermentative strain ("NNmini"). The NNmini strain exclusively produced lactate at a yield of 1.78 ± 0.2 mol

lactate/mol glucose, which is comparable to the lactate yield obtained with obligate fermentative strains lacking the terminal oxidases (1.6 mol lactate/mol glucose[12]). Upon investigating the growth phenotype of the NNmini strain, we found that it exhibited slower growth rates when it was grown under aerobic compared to anaerobic conditions, which we attribute to the lack of Pfl activity in the presence of oxygen. In these conditions, the NNmini strain has to use Pdh to synthesize acetyl-CoA, which produces additional NADH (that cannot be balanced) and is also inhibited by increased NADH concentrations commonly found under fermentative conditions. Therefore, acetyl-CoA synthesis is likely limiting aerobic growth. In the absence of oxygen, Pfl activity allows acetyl-CoA synthesis and subsequent ethanol and acetate secretion, which respectively consume more NADH and yield an extra ATP. We demonstrated that the aerobic growth impairment could be relieved by co-feeding supplements that could either be converted to acetyl-CoA (pyruvate and acetate) or replace amino acids that are derived from lower metabolism (casamino acids

(CAAs)). In the future, metabolic engineering strategies could relieve this acetyl-CoA limitation by adding modules for the intracellular production of acetyl-CoA-derived metabolites. This could be achieved by the engineering of pyruvate dehydrogenase to remove its inhibition by NADH[51], reintegration of pyruvate oxidase for quinone-dependent acetyl-CoA formation or overexpression of phosphoketolase to allow non-oxidative acetyl-CoA formation from upper metabolism. In sum, we hypothesize that the NNmini strain can be improved via various strategies in the future. Furthermore, we hypothesize that the overall resource allocation might be more optimal during anaerobic growth, during which major regulators suppress the expression of unnecessary respiratory genes[33]. For example, the strain converted to an obligate fermenting strain by deletion of the terminal oxidases required 60 days of adaptive laboratory evolution to reach growth rates equivalent to a wild-type strain growing under anaerobic conditions[12]. Therefore, we aim to perform adaptive laboratory evolution to improve the growth of the NNmini strain and will investigate the growth-improving effect through modification of global regulators of fermentative metabolism as was described before[52]. In general, understanding these systems to harness them in a predictable manner will be highly beneficial for future applications of the NNmini strain.

As a cornerstone of this work, we aimed to use the unique metabolic setup of the NNmini strain to perform re-balanced fermentations via controlled respiro-fermentative growth. As a first example, we demonstrated aerobic glycerol fermentation by genomic reintroducing *glpD*, encoding the quinone-dependent glycerol 3-phosphate dehydrogenase. Here, we observed lactate yields of $0.85 \pm 0.14$ mol lactate/mol glycerol. Notably, studies relying on microaerobic cultivations did report higher lactate yields (0.991 mol/mol glycerol), but require tightly controlled environments to maintain these[53]. Encouraged by these results, we next aimed to further push the boundaries of respiro-fermentative metabolism by replacing the native lactate synthesis route ($\Delta ldhA$) with an isobutanol production pathway (pIBA). Indeed, we could observe isobutanol production from glycerol by the NNmini + *glpD* $\Delta ldhA$ + pIBA strain under aerobic conditions. However, the observed biomass and isobutanol yields were comparably lower than for the production of lactate from glycerol with the NNmini strain. Since this might be due to the heterologous nature of the expression plasmid, we suggest that future adaptive laboratory evolution experiments could help to balance the burden from protein expression with pathway flux for optimal redox balancing and product production.

In addition to achieving the fermentative conversion of glycerol to lactate and isobutanol, the reintroduction of glycerol 3-phosphate dehydrogenase to the NNmini strain enables the generation of a proton motive force through the ETC, which is subsequently harnessed for additional ATP production. This aspect of the NNmini strain design, when considering further quinone dependent reactions, essentially permits the use of fermentation routes independent from substrate-level phosphorylation, and thus potentially paves the way for numerous innovative fermentation processes. Coherently, the potential of selectively re-balancing fermentation opens new routes, e.g. as shown for microbial glycerol valorization.

In sum, we expanded the boundaries of fermentative metabolism by creating a genetic setup that allows the modular combination of fermentative metabolism with respiratory modules. The applications of previous obligate fermentative strains were still limited by the requirement for redox-balanced substrate-product combinations. Our controlled respiro-fermentative metabolic design, however, paves the way for biochemical conversion routes using previously inaccessible substrate/product combinations.

## Methods

### Strain and plasmids construction
All *E. coli* strains used in this study are listed in Table 1. *E. coli* MG1655 derivative strain SIJ488 is carrying inducible recombinase and flippase

**Table 1 | Strains and plasmids used in this study**

| Name | Description/deletion | Reference |
|---|---|---|
| **E. coli strain** | | |
| DH5α | Cloning of overexpression constructs | |
| SIJ488 | WT carrying integrated λ-red recombinase and flippase | 54 |
| ΔubiCA | ΔubiCA | This study |
| NNQ | Δndh ΔnuoEFG ΔubiCA | This study |
| NNmini | Δndh ΔnuoEFG ΔputA ΔglcDEF Δdld Δmqo ΔdadA ΔpoxB ΔkefF ΔwrbA ΔfadE ΔyieF ΔmdaB/ygiN ΔglpD ΔgpsA ΔlldD ΔglpABC ΔsdhABCD ΔlhgO | This study |
| NNmini + glpD | Δndh ΔnuoEFG ΔputA ΔglcDEF Δdld Δmqo ΔdadA ΔpoxB ΔkefF ΔwrbA ΔfadE ΔyieF ΔmdaB/ygiN ΔlldD ΔgpsA ΔglpABC ΔsdhABCD ΔlhgO | This study |
| NNmini + glpD ΔldhA | Δndh ΔnuoEFG ΔputA ΔglcDEF Δdld Δmqo ΔdadA ΔpoxB ΔkefF ΔwrbA ΔfadE ΔyieF ΔmdaB/ygiN ΔlldD ΔgpsA ΔglpABC ΔsdhABCD ΔlhgO ΔldhA | This study |
| JW0999 | KEIO ΔputA | 56 |
| JW2121 | KEIO Δdld | |
| JW2198 | KEIO Δmqo | |
| JW1178 | KEIO ΔdadA | |
| JW0855 | KEIO ΔpoxB | |
| JW0045 | KEIO ΔkefF | |
| JW0989 | KEIO ΔwrbA | |
| JW5020 | KEIO ΔfadE | |
| JW3691 | KEIO ΔyieF | |
| JW3389 | KEIO ΔglpD | |
| JW3580 | KEIO ΔlldD | |
| JW2635 | KEIO ΔlhgO | |
| **Plasmid** | | |
| pIBA4 | Expression plasmid encoding the iBuOH pathway [pBBRori-aadA1(Sp$^R$)-T$_L$-lacI-P$_{lacIq}$] (814bb)-P$_{T7A1-O34}$-RBS$_{AS7}$-alsS$_{Bsu}$-RBS$_{AC7}$-ilvC$_{Eco-6E6}$-RBS$_{AD5}$-ilvD$_{Eco}$-RBS$_{AK7}$-kivd$_{Lla}$-RBS$_{AA2}$-adhA$_{Lla-M5}$-gfp-T$_{P22}$ | 41 |

genes[54], and was used as wild type. Gene deletions were performed by λ-Red recombineering or P1-transduction[55].

### Gene deletion via P1 transduction
Deletions of *putA*, *dld*, *mqo*, *dadA*, *poxB*, *kefF*, *wrbA*, *fadE*, *yieF*, *glpD*, *gpsA*, *lldD*, *lhgO* were generated by P1 phage transduction[55]. Donor strains were strains from the KEIO collection[56] with a kanamycin-resistance gene (KmR)[57]. The strains with the desired deletion were selected for by plating on kanamycin containing plates. Successful gene deletion was verified by determining the size of the genomic locus by PCR with DreamTaq polymerase (Thermo Scientific, Dreieich, Germany) and the respective KO-Ver primers (Supplementary Data 3). Furthermore, a PCR with DreamTaq polymerase (Thermo Scientific, Dreieich, Germany) and internal primers ("int") binding inside of the gene coding sequence was performed to confirm that no copy of the gene to be deleted was present in the genome of the transduced strain. For removal of the selective marker, a fresh culture was grown to $OD_{600}$ ~ 0.2, followed by addition of 50 mM L-rhamnose and cultivating for ~4 h at 30 °C for flippase expression induction. Colonies that only grew on LB in the absence of the respective antibiotic were isolated and successful removal of the KmR gene from the respective locus was confirmed by PCR using the locus specific KO-Ver primers and with DreamTaq polymerase (Thermo Scientific, Dreieich, Germany).

## Gene deletion by recombineering

To delete genes by recombineering, PCR with "KO" primers with 50 bp homologous overhangs and pKD4 (Addgene plasmid # 45605) as template and PrimeStar GXL polymerase (Takara Bio) was performed to generate kanamycin resistance cassettes. Cells were prepared for gene deletion by inoculating fresh cultures in LB, followed by induction of the recombinase genes by addition of 15 mM L-arabinose at OD ~ 0.4–0.5, followed by incubation for 45 min at 37 °C. The cells were harvested (11,000 × $g$, 30 s, 2 °C) and washed three times with ice cold 10% glycerol. For electroporation, ~300 ng of Km cassette PCR-product was transformed (1 mm cuvette, 1.8 kV, 25 µF, 200 Ω). Gene deletions were confirmed by selection on kanamycin containing plates and via one PCR using 'KO-Ver' primers (Supplementary Data 3) and one PCR using internal (int) primers (Supplementary Data 3), both of these being done with DreamTaq polymerase (Thermo Scientific, Dreieich, Germany). The Km cassette was removed by adding 50 mM L-rhamnose to an exponentially growing 2 mL LB culture at OD 0.5 for induction of flippase gene expression. After induction, cells were incubated for ≥3 h at 30 °C. After screening colonies for kanamycin sensitivity, removal of antibiotic resistance cassette was confirmed by PCR using 'KO-Ver' primers and DreamTaq polymerase (Thermo Scientific, Dreieich, Germany).

## Media and growth experiments

LB medium (1% NaCl, 0.5% yeast extract, 1% tryptone) was used for cloning, generation of deletion strains, and strain maintenance. When appropriate, kanamycin (25 µg/mL), ampicillin (100 µg/mL), streptomycin, (100 µg/mL), or chloramphenicol (30 µg/mL) were used. Growth experiments were carried out without antibiotics in standard M9 minimal media (50 mM Na$_2$HPO$_4$, 20 mM KH$_2$PO$_4$, 1 mM NaCl, 20 mM NH$_4$Cl, 2 mM MgSO$_4$ and 100 µM CaCl$_2$, 134 µM EDTA, 13 µM FeCl$_3$·6H$_2$O, 6.2 µM ZnCl$_2$, 0.76 µM CuCl$_2$·2H$_2$O, 0.42 µM CoCl$_2$·2H$_2$O, 1.62 µM H$_3$BO$_3$, 0.081 µM MnCl$_2$·4H$_2$O). Carbon sources were used as indicated in the text. For growth experiments, overnight cultures of engineered strains grown in LB medium containing 20 mM glucose and 10 mM acetate were incubated in 4 mL M9 medium containing 20 mM glucose and 10 mM acetate. NNmini + glpD was grown in 4 mL M9 medium with 40 mM glycerol. Cells were harvested and washed three times (6000 × $g$, 3 min) in M9 medium to remove residual carbon sources. With the washed cells, growth experiments were inoculated to an OD$_{600}$ of 0.01 in 96-well microtiter plates (Nunclon Delta Surface, Thermo Scientific) at 37 °C or 30 °C for isobutanol production experiments. Each well contained 150 µL of culture and 50 µL mineral oil (Sigma–Aldrich) to avoid evaporation while allowing gas exchange. Growth in technical triplicates was monitored at 37 °C or 30 °C in a BioTek Epoch 2 Microtiterplate reader (BioTek, Bad Friedrichshall, Germany) by absorbance measurements (600 nm) of each well every ~10 min with intermittent orbital and linear shaking. Blank measurements were subtracted and OD$_{600}$ measurements were converted to cuvette OD$_{600}$ values by multiplying with an empirically determined factor of 4.35. If not indicated otherwise, growth curves represent averages of measurements in technical duplicates and were plotted in MATLAB and python.

## Analysis of supernatants with ion chromatography

For detection of fermentation products, supernatants were analyzed by ion chromatography using the Dionex IonPac AS11-HC-4µm

Analytical/Capillary Colum (Thermo Scientific, Dreieich, Germany) with suppressed conductivity detection. Standards of D-lactate, acetate and formate were prepared with deionized water. Supernatants from cell cultures were isolated by centrifugation for 10 min at 20.238 × $g$ and transfer of the supernatant to a new Eppendorf tube. Both standards and supernatants were diluted 100-fold with water. The analytes in a 10 µL injection were separated using electrolytically generated potassium hydroxide eluent from 1 to 60 mM KOH at 0.38 mL/min within 53 min at 10 °C.

## Analysis of Δ*nuo* Δ*ndh* cultivation supernatants with IDMS

Lactate in Δ*nuo* Δ*ndh* culture supernatants was quantified using isotope dilution mass spectrometry (IDMS). The chromatographic separation was performed with 2 µL injection volume on an Agilent Infinity II 1290 HPLC system using a Kinetex EVO C18 column (150 × 2.1 mm, 3 µm particle size, 100 Å pore size, Phenomenex) connected to a guard column of similar specificity (20 × 2.1 mm, 3 µm particle size, Phenomoenex) with a constant flow rate of 0.2 mL/min with mobile phase A being 0.1% formic acid in water and phase B being 0.1% formic acid methanol (Honeywell, Morristown, New Jersey, USA) at 25 °C. The mobile phase profile consisted of the following steps and linear gradients: 0–4 min constant at 0% B; 4–6 min from 0 to 100% B; 6–7 min constant at 100% B; 7–7.1 min from 100 to 0% B; 7.1 to 12 min constant at 0% B. An Agilent 6495 ion funnel mass spectrometer was used in negative mode with an electrospray ionization source and the following conditions: ESI spray voltage 2000 V, nozzle voltage 500 V, sheath gas 250 °C at 11 L/min, nebulizer pressure 50 psig and drying gas 80 °C at 16 L/min. Compounds were identified based on their mass transition and retention time compared to standards. Chromatograms were integrated using MassHunter software (Agilent, Santa Clara, CA, USA). Absolute concentrations were calculated based on an external calibration curve and corrected for matrix effects by the use of u–$^{13}$C-lactate as internal standard. Mass transitions, collision energies, Cell accelerator voltages and Dwell times have been optimized using chemically pure standards. The parameter settings of all targets are given in Table 2.

## Analysis of glucose uptake rates in supernatants of NNmini and the Δ*nuo* Δ*ndh* strain

To determine substrate uptake rates for both strains, precultures were grown in tubes with M9 medium + 20 mM glucose. From those precultures, the main cultures with 20 mM glucose were inoculated in baffled flasks and shaken at 180 rpm. In regular intervals, the OD$_{600}$ was determined, and 1 mL culture was centrifuged for 2 min at 11,000 × $g$ to isolate the supernatants. For the NNmini strain, biological triplicates were analyzed, for the Δ*nuo* Δ*ndh* strain, technical triplicates were measured. Glucose concentrations were determined using a YSI bioanalyzer. Specific glucose uptake rates were determined according to the following formula[30]:

$$r_{Gluc}\left[mmol * g_{CDW}^{-1} * h^{-1}\right] = \mu[h^{-1}] * \left(\frac{\left(c_{Gluc,start}\left[\frac{mmol}{L}\right] - c_{Gluc,end}\left[\frac{mmol}{L}\right]\right)}{\left(c_{Bio,end}\left[\frac{g}{L}\right] - c_{Bio,start}\left[\frac{g}{L}\right]\right)}\right)$$

(1)

where $r_{Gluc}$ is the specific glucose uptake rate, $\mu$ is the strain growth rate per hour, $c_{Gluc,start}$ and $c_{Gluc,end}$ are the glucose start and end

**Table 2 | Parameter settings for IDMS measurement targets**

| Compound | Analyte type | Fragment type | Prec ion m/z (Da/eV) | Prod ion m/z (Da/eV) | Dwell time (msec) | Fragmentor voltage (V) | Collision energy (AU) | Cell accelerator (AU) | Polarity |
|---|---|---|---|---|---|---|---|---|---|
| U-$^{13}$C-Lactate | Internal standard | Quantifier | 92 | 92 | 150 | 380 | 0 | 5 | Negative |
| U-$^{13}$C-Lactate | Internal standard | Qualifier | 92 | 74 | 150 | 380 | 10 | 5 | Negative |
| Lactate | Target | Quantifier | 89 | 89 | 150 | 380 | 0 | 5 | Negative |
| Lactate | Target | Qualifier | 89 | 71 | 150 | 380 | 10 | 5 | Negative |

**Table 3 | Parameter settings for LCMS measurement targets**

| Name | Precursor ion | Product ion | Collision energy [V] | Fragmentor voltage [V] | Cell accelerator voltage [V] | Dwell time [msec] | Polarity |
|---|---|---|---|---|---|---|---|
| NAD | 664.1 | 524 | 18 | 380 | 5 | 50 | Positive |
|  |  | 428 | 26 | 380 | 5 | 50 | Positive |
| NADH | 666.1 | 649 | 17 | 380 | 5 | 50 | Positive |
|  |  | 514 | 23 | 380 | 5 | 50 | Positive |
| NADP | 744.1 | 604 | 20 | 380 | 5 | 50 | Positive |
|  |  | 508 | 32 | 380 | 5 | 50 | Positive |
| NADPH | 746.1 | 729 | 17 | 380 | 5 | 50 | Positive |
|  |  | 135.8 | 40 | 380 | 5 | 50 | Positive |
| AMP | 346 | 96 | 24 | 380 | 5 | 50 | Negative |
|  |  | 78.9 | 30 | 380 | 5 | 50 | Negative |
| ADP | 425.9 | 327.8 | 17 | 380 | 5 | 50 | Negative |
|  |  | 133.9 | 22 | 380 | 5 | 50 | Negative |
| ATP | 505.9 | 407.9 | 21 | 380 | 5 | 50 | Negative |
|  |  | 158.8 | 28 | 380 | 5 | 50 | Negative |

concentrations in mM and $c_{Bio,end}$ and $c_{Bio,start}$ is the cell dry biomass concentration in $g^{-1} L^{-1}$ at the end and start of the phase with a steady uptake rate.

## Extraction of intracellular metabolites in wild type, Δnuo Δndh strain, NNmini and NNmini + glpD

To determine intracellular metabolite concentrations, all strains were grown in 20 mM glucose or 40 mM glycerol as indicated in the respective section. During the exponential growth phase, cells equivalent to 1 mL of $OD_{600} = 1.5$ were harvested by quenching with an equal volume of −70 °C cold 70% methanol and subsequent centrifugation at 4 °C. The supernatant was discarded, and cell pellets were stored at −70 °C until proceeding with the endometabolome extraction. To extract the endometabolome, −20 °C cold chloroform and extraction fluid (50% (v/v) MeOH (LC-MS grade) and 50% TE-buffer pH 7.0 (10 mM TRIZMA, 1 mM EDTA)) were each added to the cell pellet at a volume equivalent to 300 μL per $OD_{600} = 1$. The pellet was resuspended by vortexing, followed by mixing of the sample for 2 h at −4 °C (using a rotating mixer). Subsequently, the phases were separated by centrifugation at $11,000 \times g$ at −10 °C. The upper phase was carefully isolated, filtered through a 0.22 μM filter and analyzed by LC-MS as described below. The intracellular metabolite concentrations $c_{intracellular}$ [mM] was determined as follows:

$$c_{intracellular}\,[mM] = c_{measured}\,[mM] * \left( \frac{vol_{EF}\,[\mu l]}{(vol_{EF}\,[\mu l] + biovolume\,[\mu L])} \right)^{-1} \quad (2)$$

with $c_{measured}$ [mM] being the measured concentration, $vol_{EF}[\mu L]$ being the volume of extraction fluid used, and the biovolume being calculated as follows:

$$\begin{aligned} biovolume\,[\mu L] &= CDW_{harvested}\,[mg] * cell\ specific\ volume\,[\mu L * mg^{-1}] \\ &= CDW_{harvested}\,[mg] * 2\,[\mu L * mg^{-1}] \end{aligned} \quad (3)$$

We assumed 2 μL mg⁻¹ as cell specific volume[30].

## Targeted determination of AMP, ADP, ATP, NAD, NADH, NADP and NADPH

The targets were quantified using a LC-MS/MS. Chromatographic separation was performed on an Agilent Infinity II 1290 HPLC system using a SeQuant ZIC-pHILIC column (150 × 2.1 mm, 5 μm particle size, peek coated, Merck) connected to a guard column of similar specificity (20 × 2.1 mm, 5 μm particle size, Phenomoenex) at a constant flow rate of 0.1 mL/min with mobile phases comprised of 10 mM ammonium acetate in water, pH 9, supplemented with medronic acid to a final concentration of 5 μM (phase A) and 10 mM ammonium acetate in 90:10 acetonitrile to water, pH 9, supplemented with medronic acid to a final

concentration of 5 μM (phase B) at 40 °C. The injection volume for AMP, ADP, ATP, NAD, NADP and NADPH was 2 μL. The mobile phase profile was composed of the following steps and linear gradients: 0–1 min constant at 75% B; 1–6 min from 75 to 40% B; 6 to 9 min constant at 40% B; 9–9.1 min from 40 to 75% B; 9.1 to 20 min constant at 75% B. An Agilent 6495 ion funnel mass spectrometer was used in negative and positive ionization mode with an electrospray ionization source and the following conditions: ESI spray voltage 3500 V, nozzle voltage 1000 V, sheath gas 300 °C at 9 L/min, nebulizer pressure 20 psi and drying gas 100 °C at 11 L/min. Compounds identification and chromatogram integration were performed as described for IDMS. Absolute concentrations were determined based on an external Standard curve.

Due to its low concentration, an increased injection volume of 3 μL was applied to quantify NADH. To avoid manual integration bias, a spectrum summation algorithm was used for peak integration. The determined concentrations for NADH were lower than the smallest calibration standard (10 nM), thus, NADH concentrations are "estimated concentrations" rather than calculated concentrations as is indicated in the figure legends.

Mass transitions, collision energies, Cell accelerator voltages and Dwell times have been optimized using chemically pure standards. Parameter settings of all targets are given in Table 3.

## Analysis of supernatants with high-performance anion-exchange chromatography

For the detection of alcohols and sugars at cultivation endpoints, supernatants were analyzed by high-performance anion exchange chromatography using the Dionex CarboPac MA1 anion-exchange column (Thermo Scientific, Dreieich, Germany) with pulsed amperometric detection. Standards of D-glucose, D-xylose, glycerol and ethanol were prepared with deionized water. Supernatants from cell cultures were isolated by centrifugation for 10 min at $20.238 \times g$ and transfer of the supernatant to a new Eppendorf tube. The analytes in a 10 μL injection were separated using 480 mM sodium hydroxide eluent flowing at 0.4 mL/min for 60 min.

## Sequence analysis of the NNmini strain, the NNQ strain and NNQ mutants

For whole genome sequencing, strains were grown overnight in LB medium supplemented with 20 mM glucose and 10 mM acetate. The Macherey-Nagel NucleoSpin Microbial DNA purification Kit (Macherey-Nagel, Düren, Germany) was used to extract the genomic DNA. Microbial short insert PCR-free library construction for single-nucleotide variant detection and generation of 150 bp paired-end reads on an Illumina HiSeq 3000 platform were performed by Novogene (Cambridge, UK). Breseq (Barrick Lab, Texas)[58] was used to map the obtained reads to the reference genome of *E.coli* MG1655 (GenBank

accession no. U00096.3). With the algorithms supplied by the software package, we identified single-nucleotide variants (with >50% prevalence in all mapped reads) and regions more than 2 standard coverage deviations from the global median coverage.

**Metabolite quantification of strain NNmini Δ*ldhA* +*glpD*+pIBA**

1 mL of M9 glycerol medium (with or without 20 mM acetate or 5 g L$^{-1}$ CAAs as indicated in the text) was inoculated with strain NNmini Δ*ldhA* + *glpD* + pIBA and incubated directly in a sealed GC headspace glass vial (22.5 × 46 mm, Chromtech) at 30 °C and 300 rpm for 3 days. Isobutanol quantification was performed by GC-MS/MS in SIM mode using an Agilent 5975 C inert XL EI/CI MSD system (Agilent Technologies) upgraded to MS/MS with an Evolution3 system (Chromtech) and equipped with an HP-5MS Ultra Inert column (dimensions: 30 m, 0.25 mm, 0.25 μm, Agilent Technologies) and a Combi PAL-XT auto sampler (CTC Analytics). The vial was incubated at 80 °C for 3 min and 300 μL of the head space gas were injected with a 500:1 split at 33.9 mL min$^{-1}$ to the GC-MS/MS system using a 2.5 mL syringe heated to 85 °C. The inlet temperature was set to 250 °C and a constant flow of 1 mL min$^{-1}$ helium was used as carrier gas. The oven temperature was held at 33 °C for 2 min followed by a linear temperature gradient of 25 °C min$^{-1}$ to a final temperature of 200 °C. Mass spectra were recorded starting 1.48 min after injection and *m/z* values of 55.0, 56.0, and 74.0 were used to detect isobutanol in SIM mode at 6.67 scans s$^{-1}$. Chromatograms were evaluated with the Agilent ChemStation software (Agilent Technologies). Isobutanol was quantified using a calibration curve prepared with external standards of 1 mL cultivation medium with known isobutanol concentrations measured with the same method as described above. After isobutanol quantification, the vials were opened and the medium was transferred to Eppendorf tubes and centrifuged at 13.000 × *g* for 5 min. Remaining glycerol in the supernatant was quantified using the Liquid Glycerol kit (Enzytec) according to the manufacturers' instructions. Ethanol and acetate were quantified in the supernatant using the Ethanol Assay Kit (Megazyme) and the Acetic Assay Kit (Megazyme), respectively.

### Reporting summary

Further information on research design is available in the Nature Portfolio Reporting Summary linked to this article.

## Data availability

Data supporting the findings of this work are available within the paper and its Supplementary Information files. A reporting summary for this Article is available as a Supplementary Information file. Whole-genome sequencing results for all strains were deposited on the NCBI sequencing read archive (SRA) under the BioProject number PRJNA1113154 (sample IDs: SAMN41447603 - SAMN41447608). All strains presented in the manuscript can be obtained for academic research from the corresponding author upon request. Source data are provided with this paper.

## Code availability

The CNApy software used to identify ubiquinone reducing reactions in the iML1515 model can be found at Github [https://github.com/cnapy-org/CNApy].

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

## Acknowledgements

The authors thank Ari Satanowski for fruitful discussions on the project and the manuscript. H.S-M. acknowledges funding by the Joachim Herz Foundation, the Bosch research foundation and the International Max Planck Research School "Principles of Microbial Life: From molecules to cells, from cells to interactions" (IMPRS-µLife). J.L.K. acknowledges funding by the International Max Planck Research School for Molecular

Plant Science. S.N.L. acknowledges funding from the BMBF grant Max-Kat (031B1028) and Charité Universitätsmedizin Berlin. S.K. acknowledges the support by the state of Saxony-Anhalt within the SmartProSys initiative. The authors thank Robert Landick (University of Wisconsin-Madison) for sharing the plasmid pIBA4 and Peter Claus for technical assistance with LC-MS measurements.

## Author contributions

S.N.L. and A.B.-E. conceptualized the study, S.N.L., A.B.-E., T.J.E, and S.K. supervised the work and provided feedback. S.K. and P.S. analyzed the genome-scale metabolic model with respect to quinone-reducing reactions. S.N.L., M.W., T.A., and H.S.-M. constructed the obligate fermenting strains. T.A. and B.D. evolved the NNQ strain and characterized the resulting mutants. H.S.-M. and E.M. characterized the *ΔubiCA*, NNQ and NNmini strains. J.L.K., J.N.N., O.A., H.S.-M., and N.P. characterized the *Δnuo Δndh* strain. J.L.K., O.A,. and J.N.N. compared biomass yields for acetate, pyruvate and CAA supplementation in the NNmini strain. H.S.-M. constructed the NNmini + *glpD* strain and analyzed it with J.N.N. and J.L.K. J.L.K., H.S.-M., J.N.N. and S.B. constructed and characterized the NNmini Δ*ldhA* + *glpD* + pIBA strain. H.S.-M., N.P. and S.B. performed metabolite quantifications. H.S.-M. and J.L.K. assembled data from all authors for the manuscript. H.S.-M., J.L.K., and S.N.L. drafted the original manuscript version, H.S.-M. and J.L.K wrote the manuscript with contributions from all authors.

## Funding

## Competing interests

The NNmini, NNmini + *glpD* and NNmini Δ*ldhA* + *glpD* + pIBA are part of a European patent application (EP 23 161 350.6) filed by S.N.L. with J.L.K., T.A., E.M., and H.S.-M. as inventors. The other authors declare no competing interest.
