## [Peer Review File · Nature Communications]

Engineering new-to-nature biochemical conversions by combining fermentative metabolism with respiratory modulesReviewers' Comments:

Reviewer #1:

Remarks to the Author:

The authors designed an engineered *E. coli* strain for aerobic fermentation by balancing respiration and fermentation capacity. For this, all the pathways for transferring electrons to the quinone pool were deleted, which can aerobically grow only if acetate or pyruvate was supplied due to the redox balance. Finally, the mutant strain with expression of *glpD*, which can perform respiration using glycerol as a carbon source, was modified to produce lactate and isobutanol as fermentative production. The idea seems novel, which are supported by the experimental data. I would like to point the following points for improving manuscript.

Comments

#1. The goal of this study is to prevent the use of oxygen as a terminal electron acceptor without compromising ETC. Therefore, the changes of intracellular ATP and NADH levels are considered to be important. Adding the measurement data for ATP and NADH/NAD⁺ would increase the credibility of the argument, for example supply NADH/NAD⁺ data in line 199 to 202 and/or ATP concentration in line 396.

#2. In line 313, referring to Supplement Figure 4, in a situation where cell growth is inhibited due to poor conversion of acetyl-CoA from pyruvate, growth appears to be restored when pyruvate is supplied. It seems that further explanation is needed.

#3. In the absence of NADH dehydrogenase, ATP is produced only through substrate-level phosphorylation. But using acetate as a carbon source, additional ATP is needed, is there any problem with this?

#4. Considering both Fig. 5B and Fig. 6B are results from the cells growth with glycerol as a carbon source, why are the growth so different? (60 h and 200 h).

#5. "yield" is used interchangeably with g/g and mol/mol. Unifying all of these in the text to mol/mol would make the readers easy to understand.

#6. It is better to explain the picture numbers in the order of the text. Figure 4D should be after 4A.

#7. As mentioned in the Discussion part, it seems more valuable to produce fermentative chemicals as a single carbon source by introducing an additional Acetyl-CoA production pathway rather than an additional acetate supply. Could you explain further about the metabolic strategy?

Reviewer #2:

Remarks to the Author:

The authors propose a clever and practical strategy to employ respiratory modules for redox balancing of newly designed fermentative pathways in *Escherichia coli*. Their approach allows to feed electrons into the quinone pool from selected single redox reactions, thus enabling targeted oxidation of excess reducing equivalents when necessary, and reach redox-balanced aerobic fermentations without a need to fine-tune oxygen supply. They demonstrate viability of their approach by constructing strains that ferment glycerol to lactate and glycerol to isobutanol. The manuscript is well written and the described methodology represents general interest for researchers in metabolic engineering of bacterial producers.

A few remarks:

- There seems to be a contradiction between the observed low aerobic growth rate in the NNmini strain and the general microbial "rate-yield tradeoff" concept, stating that fermentative growth causes a decrease of biomass yield, while at the same time, an increase in substrate consumption and specific growth rate (see e.g., the reference [47] in the manuscript). The authors should slightly expand their Discussion part on this topic, pointing out the putative reasons for this discrepancy;
- What were the substrate uptake rates? How did glucose consumption in NNmini compare to that of wild-type? Substrate uptake time-course should be presented in the respective Figures together with growth curves;
- The scheme presented in Figs. 2B, 5A and 6A, showing the modifications of redox equivalent fluxes, is a bit misleading. The scheme depicts all electrons that reach quinone pool as coming via NAD(P)H. However, not all ubiquinone reduction reactions listed in Fig. 2A involve NAD(P)H, see e.g., *sdh*, *fadE*;
- The term "respiro-fermentative metabolism" is widely used, for example regarding Crabtree-positive yeast, but not originally "coined" by authors to denote just this particular growth phenotype (as stated in lines 173-174). Here something like "controlled (or regulable, or directed,...etc) respiro-fermentative metabolism" might be a more distinctive term.

Point-by-point response to the reviewer comments

Reviewer #1 (Remarks to the Author):

The authors designed an engineered *E. coli* strain for aerobic fermentation by balancing respiration and fermentation capacity. For this, all the pathways for transferring electrons to the quinone pool were deleted, which can aerobically grow only if acetate or pyruvate was supplied due to the redox balance. Finally, the mutant strain with expression of *glpD*, which can perform respiration using glycerol as a carbon source, was modified to produce lactate and isobutanol as fermentative production. The idea seems novel, which are supported by the experimental data. I would like to point the following points for improving manuscript.

We thank the reviewer for the positive feedback.

Comments

#1. The goal of this study is to prevent the use of oxygen as a terminal electron acceptor without compromising ETC. Therefore, the changes of intracellular ATP and NADH levels are considered to be important. Adding the measurement data for ATP and NADH/NAD⁺ would increase the credibility of the argument, for example supply NADH/NAD⁺ data in line 199 to 202 and/or ATP concentration in line 396.

We thank the reviewer pointing this out and agree that those experiments are valuable additions to the manuscript. We quantified the intracellular NADH and NAD⁺ concentrations for our glucose growing strains and added the results in Supplementary Figures 1 and 2, and refer to them as follows in the main text:

Lines 206– 213: *“Indeed, the specific glucose uptake rate during this growth phase was 8.42 mmol gCDW⁻¹ h⁻¹ (Supplementary Fig. 1C), which matches previous reports for the aerobic specific glucose uptake rate of a wild-type strain (7.62 ± 0.43 mmol gCDW⁻¹ h⁻¹)³¹. Upon quantifying the NADH/NAD⁺ ratio of cells growing with 20 mM glucose, we found a slightly increased ratio for the Δ*ndh* Δ*nuoEFG* strain (Supplementary Fig. 1D-1F), which indicates metabolic perturbations by the NADH dehydrogenase deletions. To further push metabolism towards fermentative growth, we proceeded to delete the other identified genes.”*

Lines 246 – 250: *“Furthermore, the NADH/NAD⁺ ratio of the NNMini strain was increased ~16-fold compared to the wildtype, which is in line with previous reports of ~10-fold^{32,33} increased NADH/NAD⁺ ratios of fermentatively grown wildtype cells compared to aerobically grown ones (Supplementary Fig. 2B-2D).”*

As suggested by the reviewer, we also quantified intracellular ATP levels of glucose and glycerol growing strains and added the result in Supplementary Figure 6B, which we refer to as follows in the main text (lines 412 – 420):

*“This phenotype is likely due to respiratory ATP production in the ETC coming from the reoxidation of the quinol (QH₂) that is generated by *GlpD* (Fig. 1A). Measurements of intracellular ATP concentrations supported this hypothesis. Here, ATP concentrations of the NNmini +*glpD* strain grown on 40 mM glycerol were ~2.6 fold higher than those of the NNmini strain grown on 20 mM glucose (Supplementary Fig. 6B). Notably, ATP concentrations in both strains were still lower than in the Δ*nuo* Δ*ndh* strain grown on 20 mM glucose or 40 mM glycerol, respectively. Thus, ATP synthesis appears to be reduced (Supplementary Fig. 6B), which is in line with our expectation of reduced respiratory chain activity.”*

#2. In line 313, referring to Supplement Figure 4, in a situation where cell growth is inhibited due to poor conversion of acetyl-CoA from pyruvate, growth appears to be restored when pyruvate is supplied. It seems that further explanation is needed.

We thank the reviewer for the comment and expanded the explanation on why pyruvate might improve growth to some extent as follows (lines 335 – 343):

*“In case of pyruvate supplementation, converting the supplied pyruvate to acetyl-CoA via *Pdh* results in the formation of one NADH. However, converting 1 molecule of pyruvate to 1 molecule of lactate also consumes one NADH. Each additional conversion of supplied pyruvate to lactate should further lower intracellular NADH levels, thus reducing *Pdh* inhibition. Then, a fraction of the supplied pyruvate can be used for acetyl-CoA synthesis via *Pdh*, which should improve growth at higher concentrations of supplied pyruvate. Indeed, biomass yields and growth rates of the NNmini strain supplemented with pyruvate were lower compared to acetate or CAAs (max. OD = 0.47 with 8 mM pyruvate, growth rate = 0.108 ± 0.003 h⁻¹).”*

#3. In the absence of NADH dehydrogenase, ATP is produced only through substrate-level phosphorylation. But using acetate as a carbon source, additional ATP is needed, is there any problem with this?

Indeed, at least one additional ATP is consumed to convert acetate to acetyl-CoA. However, our findings indicate that growth is significantly improved indicating that acetyl-CoA rather than ATP is limiting growth. We expanded the relevant section to more clearly state our observations (lines 326 – 330):

“In case of acetate supplementation, despite the cost of 1 ATP for acetyl-CoA formation, growth was significantly improved (maximum $OD_{600} = 0.8$ with 4 mM acetate, max. growth rate = 0.277 h^{-1}). This hints that acetyl-CoA limits growth of the NNmini strain rather than ATP yields from substrate-level phosphorylation.”

#4. Considering both Fig. 5B and Fig. 6B are results from the cells growth with glycerol as a carbon source, why are the growth so different? (60 h and 200 h).

The reviewer correctly points out large growth differences between the lactate-producing strain shown in figure 5B and the isobutanol producing strain shown in Fig. 6B. The lactate synthesis route is part of the native *E. coli* metabolism and therefore likely fine-tuned for optimal growth. Our current hypothesis is optimizing the expression levels of the heterologously expressed isobutanol production enzymes will likely improve strain growth in the future. To point this out more clearly, we modified lines 459 – 467 as follows:

*“While growth on glycerol could be achieved, the strain grew much slower compared to its lactate producing ancestor. This differences is potentially attributed to the associated protein and plasmid burden as the plasmid was optimized for a different genetic background³⁷. Additionally, redox factor recycling requires the activity of 3 heterologous (*alsS*, *kivd* and *adhA*) and 2 native genes (*ilvD* and *ilvC*) for isobutanol production compared to only 1 gene when lactate is the fermentation product (*ldhA*). In the future, performing enzyme expression optimization as it was carried out in the original paper presenting the pIBA plasmid together with adaptive laboratory evolution could be used to improve pathway activity and thereby growth.”*

#5. "yield" is used interchangeably with g/g and mol/mol. Unifying all of these in the text to mol/mol would make the readers easy to understand.

We agree with the reviewer and unified everything to mol/mol.

#6. It is better to explain the picture numbers in the order of the text. Figure 4D should be after 4A.

We thank the reviewer for pointing this out and switched the figure order accordingly.

#7. As mentioned in the Discussion part, it seems more valuable to produce fermentative chemicals as a single carbon source by introducing an additional Acetyl-CoA production pathway rather than an additional acetate supply. Could you explain further about the metabolic strategy?

We agree with the reviewer that adding acetyl-CoA forming modules is an exciting strategy to optimize the fermenting strains in the future. We expanded our thoughts on this in lines 521 - 526:

“In the future, metabolic engineering strategies could relieve this acetyl-CoA limitation by adding modules for the intracellular production of acetyl-CoA-derived metabolites. This could be achieved by the engineering of pyruvate dehydrogenase to remove its inhibition by NADH, reintegration of pyruvate oxidase for quinone-dependent acetyl-CoA formation or overexpression of phosphoketolase to allow non-oxidative acetyl-CoA formation from upper metabolism.”

Reviewer #2 (Remarks to the Author):

The authors propose a clever and practical strategy to employ respiratory modules for redox balancing of newly designed fermentative pathways in *Escherichia coli*. Their approach allows to feed electrons into the quinone pool from selected single redox reactions, thus enabling targeted oxidation of excess reducing equivalents when necessary, and reach redox-balanced aerobic fermentations without a need to fine-tune oxygen supply. They demonstrate viability of their approach by constructing strains that ferment glycerol to lactate and glycerol to isobutanol. The manuscript is well written and the described methodology represents general interest for researchers in metabolic engineering of bacterial producers.

We thank the reviewer for the nice words and the support.

A few remarks:

- There seems to be a contradiction between the observed low aerobic growth rate in the NNmini strain and the general microbial “rate-yield tradeoff” concept, stating that fermentative growth causes a decrease of biomass yield, while at the same time, an increase in substrate consumption and specific growth rate (see e.g., the reference [47] in the manuscript). The authors should slightly expand their Discussion part on this topic, pointing out the putative reasons for this discrepancy;

We agree with the reviewer that this section needed some clearer explanations. First of all, we now determined that the specific glucose uptake rate of the NNMini strain under aerobic conditions indeed matches that of a fermentatively growing wild type (reported in lines 206-212, see answer to the next comment). However, we believe

that the mentioned “rate-yield tradeoff” concept cannot be used here because redox balancing issues (rather than resource allocation constraints) limit the growth of the NNmini strain: the absence of Pfl in aerobic conditions results in limited acetyl-CoA availability and lower ATP yield, thus causing the reduced growth rates in aerobic conditions. We re-phrased the section to explain this more clearly (lines 508 – 518):

“Upon investigating the growth phenotype of the NNmini strain, we found that it exhibited slower growth rates when it was grown under aerobic compared to anaerobic conditions, which we attribute to the lack of Pfl activity in presence of oxygen. In these conditions, the NNmini strain has to use Pdh to synthesize acetyl-CoA, which produces additional NADH (that cannot be balanced) and is also inhibited by increased NADH concentrations commonly found under fermentative conditions. Therefore, acetyl-CoA synthesis is likely limiting aerobic growth. In the absence of oxygen, Pfl activity allows acetyl-CoA synthesis and subsequent ethanol and acetate secretion, which respectively consume more NADH and yield an extra ATP. We demonstrated that the aerobic growth impairment could be relieved by co-feeding supplements that could either be converted to acetyl-CoA (pyruvate and acetate) or replace amino acids that are derived from lower metabolism (casamino acids (CAAs)).”

- What were the substrate uptake rates? How did glucose consumption in NNmini compare to that of wild-type? Substrate uptake time-course should be presented in the respective Figures together with growth curves;

We thank the reviewer the comment and determined the substrate uptake rates for the $\Delta nuoEFG \Delta ndh$ and the NNmini strains on 20 mM glucose, and compare the results to a previous study by the co-author Simon Boecker (doi: [10.15252/msb.202110504](https://doi.org/10.15252/msb.202110504)) characterizing aerobic and fermentative glucose uptake rates for an *E. coli* MG1655 wild type (which our strains are derived from). We added the results in Supplementary Figures 1 and 2 and refer to them as follows in the main text:

Lines 207 – 209: *“Indeed, the specific glucose uptake rate during this growth phase was $8.42 \text{ mmol gCDW}^{-1} \text{ h}^{-1}$ (Supplementary Fig. 1C), which matches previous reports for the aerobic specific glucose uptake rate of a wild-type strain ($7.62 \pm 0.43 \text{ mmol gCDW}^{-1} \text{ h}^{-1}$)²⁹.”*

Lines 239 – 242: *“In line with this, the specific substrate uptake rate of the NNmini strain ($14.19 \pm 1.54 \text{ mmol gCDW}^{-1} \text{ h}^{-1}$, Supplementary Fig. 2A) was comparable to the previously reported specific glucose uptake rate of a fermentatively growing wild type ($13.48 \pm 0.04 \text{ mmol gCDW}^{-1} \text{ h}^{-1}$)²⁹.”*

- The scheme presented in Figs. 2B, 5A and 6A, showing the modifications of redox equivalent fluxes, is a bit misleading. The scheme depicts all electrons that reach quinone pool as coming via NAD(P)H. However, not all ubiquinone reduction reactions listed in Fig. 2A involve NAD(P)H, see e.g., *sdh*, *fadE*;

We agree with the comment and changed the respective schemes to also depict direct electron transfer to quinone pool.

- The term “respiro-fermentative metabolism” is widely used, for example regarding Crabtree-positive yeast, but not originally “coined” by authors to denote just this particular growth phenotype (as stated in lines 173-174). Here something like “controlled (or regulable, or directed,...etc) respiro-fermentative metabolism” might be a more distinctive term.

We thank the reviewer very much for drawing our attention to this nomenclature from yeast. We now cite the relevant paper in the text, refer to “controlled respiro-fermentation” in the entire manuscript and explain our wording in lines 174 – 177:

“We called this growth phenotype controlled respiro-fermentative growth to reflect the combination of respiratory and fermentative growth but to differ from natural, uncontrolled respiro-fermentative metabolism (“crabtree effect”).”

Reviewers' Comments:

Reviewer #1:

Remarks to the Author:

The authors provided the valuable data for the substrate consumption rate as well as intracellular NADH, NAD⁺, and ATP concentrations. The idea they suggested is more clearly demonstrated with those data and the manuscript is now acceptable.

Reviewer #2:

Remarks to the Author:

The authors have addressed all my comments. I recommend publication.